# Knowledge, Attitudes and Practices Regarding Antibiotic Use and Antibiotic Resistance: A Latent Class Analysis of a Romanian Population

**DOI:** 10.3390/ijerph19127263

**Published:** 2022-06-14

**Authors:** Elena Narcisa Pogurschi, Carmen Daniela Petcu, Alexandru Eugeniu Mizeranschi, Corina Aurelia Zugravu, Daniela Cirnatu, Ioan Pet, Oana-Mărgărita Ghimpețeanu

**Affiliations:** 1Public Health and Food Safety Laboratory, Department Formative Science in Animal Breeding and Food Industry, Faculty of Animal Productions Engineering and Management, University of Agronomic Sciences and Veterinary Medicine of Bucharest, 011464 Bucharest, Romania; elena.pogurschi@usamv.ro; 2Department Animal Production and Public Health, Faculty of Veterinary Medicine, University of Agronomic Sciences and Veterinary Medicine of Bucharest, 050097 Bucharest, Romania; carmen28petcu@gmail.com; 3Research and Development Station for Bovine, 310059 Arad, Romania; alexandru.mizeranschi@scdcbarad.ro; 4Department-Fundamental Disciplines, Faculty of Midwifery and Nursing, University of Medicine and Pharmacy “Carol Davila”, 050474 Bucharest, Romania; dr_corinazugravu@yahoo.com; 5Department of Pharmaceutical Sciences, Faculty of Pharmacy, “Vasile Goldis” Western University of Arad, 310025 Arad, Romania; daniela.cirnatu@insp.gov.ro; 6Department of Biotechnologies, Bioengineering, Faculty of Animal Resources, Banat University of Agricultural Science and Veterinary Medicine “Regele Mihai I al Romaniei”, 300645 Timisoara, Romania; ioanpet@usab-tm.ro

**Keywords:** antibiotics, antimicrobial resistance, knowledge, attitudes, practices

## Abstract

Considering the major limitations of the latest studies conducted in Romania on the knowledge, attitudes and practices (KAPs) of antibiotic use and antibiotic resistance, we conducted this study to assess this major public health threat. A cross-sectional survey based on a validated questionnaire was conducted among the general population of Romania for a period of 5 months, i.e., September 2021–January 2022. The questionnaire was distributed using Google Form and it covered demographic characteristics and KAP assessments consisting of 12 items on knowledge, 10 items on attitudes and 3 items on practices. Latent class analyses (LCAs) were conducted to group respondents based on their responses. The response rate was 77%, of which females responded in a greater number (*n* = 1251) compared to males (*n* = 674). For most of the respondents (67.32%, *n* = 1296), the education level was high school, while 23.58% (*n* = 454) of respondents were college graduates. One in three Romanians (33.3%) know the WHO predictions related to this topic. Overall, the Romanian population is less disciplined when it comes to completing antibiotic treatments, as 29.19% of the respondents stop the course of antibiotic administration if their symptoms improve. The key findings from the present study may help policy makers in designing targeted interventions to decrease confusion, ambiguity or misconceptions about antibiotic use.

## 1. Introduction

Antibiotics and their use in the fight against infectious diseases in both humans and animals represents one of the greatest discoveries of the 20th century [1,2]. However, it became a worrying threat to the public health a century later, with many social and economic implications. The resistance of microorganisms to antibiotics naturally occurs over time through genetic mechanisms. In this context, since the last two decades have not been marked by the discovery of new antibacterial substances [3], the phenomenon of antimicrobial resistance (AMR) has intensified, generating substantial costs for the healthcare systems around the world [4]. Simultaneously, the excessive and inadequate use of antibiotics contribute to this concern that hovers over the global health system, already affected by the COVID-19 crisis [5].

The consumption of animal products obtained from animals treated with antimicrobial substances also contributes to the development and spread of AMR [6]. Even though antibiotics have been banned from feeding farm animals, many animal products still contain antibiotic residues. Meat seems to be, by far, the most contaminated animal product with antibiotic residues [7]. The presence of antibiotics has been reported in poultry meat [8,9,10,11,12], pork [10,13] and most commonly in beef [8,14,15,16,17,18]. The contamination of milk [19,20,21,22,23], fish and seafood [24,25,26,27,28], and even eggs [29,30,31], with antibiotic residues has also been reported in numerous studies. Although they are valuable nutritional contributors to our diet, animal products have been shown, in many cases, to increase consumers’ exposure to antibiotics. In this context, antibiotic residues in animal foods have become a serious public health problem.

In addition to food, the general population is exposed to antibiotics through various treatments throughout their lifetime. In general, penicillins, fluoroquinolones and macrolides are most commonly prescribed in humans, whereas in animals, the main antibiotics are tetracyclines and sulfonamides. In addition to the many side effects of these substances, AMR must be considered, as it leads to numerous annual deaths and increased public health costs. Raising public awareness of the side effects of antibiotics can prevent the spread of AMR and its effects, significantly reducing the costs to health systems in the uncertain future of human health. Even though maximum limits have been set for antibiotics in food, in the long term, the intake of antibiotics, even in quantities that do not exceed the maximum limits, can lead to the development of AMR. The excessive use of antibiotics should therefore be avoided by both veterinarians and human doctors, but also by the general population. The reduction in AMR should include measures to educate the public on the efficacy of these substances and their proper use, as demonstrated by numerous studies [32,33,34,35,36,37,38,39,40,41,42,43,44,45]. Investigating the public knowledge, attitudes and behaviors towards antibiotic use contributes to the organization of effective campaigns to prevent AMR. By analyzing the practices of using antibiotics and correcting the inappropriate ones, we can act strategically and punctually to prevent the development of antibiotic resistance.

To the best of our knowledge, no study has been conducted in Romania to highlight the aspects related to antibiotic use and knowledge in the general population. The latest studies conducted in Romania on the knowledge, attitudes and practices (KAPs) of antibiotic use and antibiotic resistance have had major limitations. One of these limits is the regional level at which the studies were conducted. Studies were set up in a single region—the eastern part of Romania [46], or in a single county—more precisely, in Mureș County [47]. Moreover, the respondents involved had a certain level of education, such as students [48] or resident doctors, or had a direct need for medication, being interviewed at the pharmacy while purchasing various drugs [46].

Thus, this study is designed to determine KAPs towards antibiotic use among the general public in Romania, as well as to provide important information about the population’s perceptions towards antibiotic resistance.

## 2. Materials and Methods

### 2.1. Study Design and Sample Selection

A cross-sectional survey based on a validated questionnaire was conducted among the general population of Romania from September 2021 to January 2022 inclusive. The questionnaires used in previous studies [38,42,44,49,50], as well as the survey conducted in the Special Eurobarometer 445: AMR [51] led us to the development of an effective questionnaire that provided important information about the Romanian population’s perception regarding the KAPs of antibiotic use and antibiotic resistance. For this study, 2500 adults from the 4 macro-regions of Romania were invited to access the questionnaire. Of the 2500 invited participants, only 2067 (82.7%) completed the questionnaire. The ambiguous or incomplete answers were eliminated so that, in the end, the statistical data processing was based on 1925 remaining individual answers.

### 2.2. The Questionnaire

The questionnaire included two parts: 1. Demographic characteristics and 2. KAPs of antibiotic use and antibiotic resistance. The first part of the questionnaire included questions about demographic characteristics, such as gender, age, educational level, civil status and income. The second part included three sections that aimed to evaluate the Romanian population’s perception regarding the KAPs of antibiotic use and antibiotic resistance. The first section consisted of 12 statements aimed to evaluate the knowledge regarding antibiotic efficacy, mode of action and antibiotic resistance (AK01–AK12). Attitudes regarding antibiotic consumption were assessed in the second section of the questionnaire, which included 10 questions (AC01–AC10). The last section of the questionnaire contained 3 questions (AU01–AU03) that aimed to identify the respondents’ antibiotic use practices. The questionnaire was prepared in Romanian, and then translated into English. The invited participants filled in the questionnaire only in Romanian, so that, for each questionnaire, 15 min were sufficient to complete it.

### 2.3. Data Analysis

The data were first inspected and cleaned using Microsoft Excel 2016. From the original dataset comprised of 2067 answers, entries that were ambiguous (e.g., multiple answers for questions regarding personal details or questions with “agree”/“unsure”/“disagree” choices) were discarded. The final dataset contained 1925 individual answers. All data analysis steps were performed in the R programming environment v.4.1.2 [52]. Data were read from Excel into R and descriptive statistics were computed using the packages readxl v.1.3.1 and dplyr v.1.0.8, respectively, from the tidyverse package collection [53]. Latent Class Analysis (LCA) was performed with the poLCA package v.1.4.1 [54] for all the different question groups, except for questions involving personal details. In all cases, respondents were grouped into two classes. First, the three questions from the Antibiotic Use (AU) practices group were analyzed together, according to respondents’ similarity in their answer patterns. Original answers, including multiple-choice entries, were analyzed without any prior editing. For the remaining two sets of questions, answers were relabeled according to whether they represented the correct knowledge or attitude related to antibiotics. Correct answers were labeled as “1” and incorrect answers as “3”. As such, for questions where choice “a” (“agree”) was the correct response, the “a” choice was labeled as “1” and the “c” choice (“disagree”) was re-labeled as “3”. The opposite was performed for questions where choice “c” was the correct answer. The “b” choice (“unsure”) was always represented as “2”. Following this process of re-labeling, LCA was used in order to classify respondents in 2 classes, according to the similarity of response patterns. This was performed for two sets of questions: (a) the 12 questions belonging to the AK (antibiotic and antibiotic resistance knowledge) group and (b) the 10 questions from the AC (antibiotic consumption attitudes) group. The answer of choice “1” was used as reference when computing odds ratios (Ors) for the outcome variables based on LCA class membership.

### 2.4. Statistical Modeling of KAPs of Antibiotic Use and Antibiotic Resistance

For the three KAPs studied traits, LCA class 1 comprised the respondents with the largest proportion of appropriate KAPs regarding antibiotics, whereas respondents from class 2 had a majority of answers demonstrating incorrect KAPs, or a lack of certainty (as signified by the answer “not sure”). Associations of respondents’ personal details with the antibiotic-related traits of focus in this study were evaluated via logistic regression at a significance level threshold of *p* < 0.05, with the answers to questions regarding personal details modeled as categorical predictors. The base R function glm was used to implement logistic regression. Results are reported as odds ratios (ORs) and 95% confidence intervals (CIs). The membership in the previously described LCA classes (classes 1 and 2) for each of the traits was used as a dependent variable in logistic regression, with class 1 used as reference for expressing the OR. Independent variables included the following demographic traits: age (18–24, 25–34, 35–44, 45–54, 55–64 and >65 years old), sex (male, female), level of education (secondary, high school, graduate and postgraduate level), civil status (married, in a relationship, single) and income level (<1500 RON, 1501–3000 RON, 3001–4500 RON, >4500 RON, undisclosed).

## 3. Results

### 3.1. Response Rates

Of the 2500 people invited to participate in this study, only 82.67% accessed the questionnaire link and 77% (*n* = 1925) correctly completed the proposed questionnaire. Of the participants, 64.99% (*n* = 1251) were female and 35.01% were male. As in many other studies [55,56,57], the tendency of women to respond in greater numbers is also evident in this study. The majority of the participants, 69.19% (*n* = 1332), were young and aged between 18 and 24 years. According to the acquired data (Table 1), a decreasing distribution of respondents can be observed in terms of age, the lowest percentage (0.26%) of respondents belonging to the age group of over 60 years. The tendency for young people to participate in more numbers than older people is found in other studies with the same purpose [45,58,59,60,61]. Furthermore, 67.32% (*n* = 1296) reported that their level of education was high school and 23.58% (*n* = 454) were college graduates; only 16.05% (*n* = 309) were married and 37.09% considered the monthly income to be confidential.

### 3.2. Knowledge Related to Antibiotics Efficacy, Mode of Action and Antibiotic Resistance

Ten statements assessed the knowledge related to the efficacy and mode of action of antibiotics, and two statements assessed the knowledge about antibiotic resistance (Table 2).

Knowledge was considered satisfactory if ≥80% of the participants answered correctly for each statement. Only 22.23% of the respondents correctly agreed on the statement “Antibiotics kill bacteria” and almost half, 49.19%, to the statement “Antibiotics treat the majority of diseases”. While the question might seem ambiguous, its purpose was to gradually depict the general knowledge of the population regarding antibiotic use, followed by more specific questions. In this context, the confusion about whether or not antibiotics treat the majority of diseases was apparent in this survey, since a quarter of participants (24.31%) were unsure about this subject. There was also a great confusion about the effectiveness of antibiotics on viruses, since only 32.88% correctly disagreed on the statement “Antibiotics treat viral infections”, and 37.51% believed that antibiotics treat viral infections and 29.61% were unsure. Furthermore, the majority of respondents incorrectly agreed on three statements regarding the effectiveness of antibiotics: “Antibiotics reduce pain and inflammation” (52.31%), “Antibiotics significantly reduce fever” (47.27%) and “Antibiotics help a lot with dental pain” (47.12%), and only one third disagreed (31.17%, 30.18% and 31.22%). The respondents unsure of these statements did not exceed 25%. Over 1/2 of respondents (53.87%) responded correctly to the statement “Excessive antibiotic use leads to AMR”, while those unsure or those who did not agree with this statement were 24.31% and 21.82%, respectively. Of the respondents, 73.77% and 81.77% correctly stated that antibiotics can cause allergies and that antibiotics have side effects, respectively. A considerable percentage (57.35%) was aware that antibiotic treatment should not be stopped when symptoms have disappeared. However, 24.99% presented an incorrect answer for this statement, which is quite worrying in terms of the phenomenon of increased AMR. Respondents were asked to present their opinion on some information about antibiotic resistance and the severity of the phenomenon. The percentage of respondents who agreed with the statement “Animal products can contain antibiotic residues, which may increase AMR when entering the human body” was less than half, specifically 40.57%, while 47.95% were unsure and 11.48% disagreed. The data presented in Table 2 show that 33.3% of respondents agree with the WHO predictions [57] on the severity of antibiotic-resistant microorganisms. In addition, a reduced percentage of respondents (4.52%) did not agree with these forecasts, while the largest percentage of respondents (62.18%) was uncertain. Regarding LCA, respondents who provided more accurate knowledge were grouped into LCA class 1, which comprised 46.7% (*n* = 899), while the percentage of people who demonstrated less accurate knowledge or uncertainty (LCA class 2) was 53.5% (Figure 1).

In terms of age, older people were, in general, less likely to belong to class 2 (less accurate knowledge) vs. class 1 (more accurate knowledge), compared to people from the 18–24 age group. The most significant difference was observed in the 55–64 age group, which was approximately 4 times less likely (OR = 0.244, CI = 0.068–0.689, *p*-value = 0.014) to have inaccurate knowledge of antibiotics compared to people between 18–24 years old. Similarly, women were approximately twice less likely (OR = 0.57, CI = 0.465–0.699, *p*-value < 0.001) than men to have inaccurate vs. accurate knowledge of antibiotics and antibiotic resistance, i.e., to belong to LCA class 2 vs. class 1. In regard to their education, higher levels of education were significantly associated with a reduced likelihood of having inaccurate knowledge about antibiotics and antibiotic resistance. People who had a postgraduate education were approximately 10 times less likely (OR = 0.092, CI = 0.02–0.305, *p*-value < 0.001) to have inaccurate knowledge compared to people who only had a secondary education. There were no significant differences noticed in terms of civil status, whereas for income level, the only statistically significant difference was observed between people earning RON 1501–3000 and those earning <RON 1500, with the former having an increased likelihood (OR = 1.356, CI = 1.02–1.804, *p*-value = 0.036) to belong to LCA class 2 vs. class 1, i.e., to have less accurate knowledge about antibiotics and antibiotic resistance on average (Table 3).

### 3.3. Attitudes towards Antibiotic Consumption

Ten statements assessed attitudes toward antibiotic consumption (Table 4).

Unexpectedly, over half of the respondents (52.73%) incorrectly agreed with the statement ″I always try to have an antibiotic in the house″, which is considered a negative attitude. The majority (83.64%) had an appropriate attitude disagreeing with the statement ″If a family member feels unwell, I usually give them an antibiotic″, whereas 8.42% incorrectly agreed with this statement. The percentage of respondents who confidently used remaining antibiotics for reoccurring similar symptoms was 21.19%, while the vast majority (67.58%) disagreed. A total of 83.32% of the respondents agreed with the statement ″I use antibiotics according to the instructions for use″. Over half of the respondents (59.43%) disagreed with the statement ″I most often use antibiotics when the symptoms are cold/flu″, which is an appropriate attitude regarding antibiotics consumption. Similarly, 54.23% of respondents showed a correct attitude towards the statement ″I most often use antibiotics when the symptoms are related to toothache″, by disagreeing. Even so, almost 30% of the population participating in the present study agreed with these two statements, demonstrating a misguided attitude towards antibiotic consumption. Only 9.19% agreed with statement ″I most often use antibiotics when the symptoms are related to gastrointestinal disorders″, which means that the use of antibiotics is associated more often with toothaches and colds than with gastrointestinal diseases in the respondent population. Similar percentages of respondents gave appropriate answers to a question that appeared reformulated in both the knowledge and attitude sections, which suggests that respondents who had adequate knowledge of antibiotics were more likely to also have a positive attitude toward the use of antibiotics. If 57.35% correctly disagreed with the AK10 statement from the knowledge section, 58.96% of respondents disagreed with the same statement, which was reformulated as AC09 in the attitude section. The combination of drug–food (AC10), in this case antibiotic–food, is of particular importance to 57.3% of respondents. Moreover, 32% were unsure if the statement ″There are many interactions between antibiotics and food consumed″ was correct, while 10.7% disagreed. Among the demographic variables included in this study, the likelihood of respondents to belong to class 2 vs. class 1 in terms of attitudes towards antibiotic consumption was significantly associated with all five traits (Table 3). Class 2 represented the population with more improper attitudes towards antibiotic consumption and comprised 589 respondents (30.6%), while the percentage of people who showed proper attitudes was more than two-fold (69.4%, *n* = 1336, Figure 2).

People of ages 25–34 years were half as likely (OR = 0.489; CI = 0.344–0.689; *p*-value < 0.001) to have improper attitudes about antibiotic consumption rather than proper attitudes (i.e., to belong to LCA class 2 instead of class 1), compared to people of ages 18–24 years. A similar but stronger trend was noticed for people aged 35–44 and 45–54 years, where OR values of 0.256 and 0.227 were observed, respectively. Similar results were observed for gender, with females being less likely to have improper attitudes compared to males, and for education, where people with higher levels of education were less likely to have proper attitudes compared to those with the lowest level of education. Regarding civil status, single people were approximately twice as likely (OR = 2.155; CI = 1.37–3.451; *p*-value = 0.001) than married people to belong to class 2. Surprisingly, a similar trend was observed for levels of income, with people with higher levels of income (as well as an undisclosed income) being more likely to have improper attitudes towards antibiotic consumption than people with the lowest income level.

### 3.4. Practices Regarding Antibiotic Consumption

For the question “How do you use antibiotics?”, the most frequent answer at 78.86% (*n* = 1518) was “only on the doctor’s prescription” (Table 5).

The pharmacist’s advice only influenced 3.84% (*n* = 74) of the respondents. The percentage of respondents who answered that they used an antibiotic when they fell sick, with or without prescription, was 10.29% (*n* = 198). The experience of friends/acquaintances in the use of antibiotics influenced the way to use such a drug only for a small number of respondents (*n* = 135). Regarding the first impulse when they fell sick, more than half of respondents (53.82%) said they went to see a doctor. Almost one-third (29.92%) reported that they were informed of the symptoms on the Internet. Less than 10% of respondents (8.52%) said they were based on their own experience when they feel sick, which implied self-medication. The pharmacist’s advice was sought by the fewest study participants when they felt unwell (7.74%). Among the respondents, 71.79% first went to the doctor following symptoms of a disease and only then used antibiotics. Only 10.39% (*n* = 200) of respondents considered that the passage of time did not help at all. This population adopted an inappropriate practice of using antibiotics when the symptoms of the disease appeared. They declared that, immediately after the onset of the symptoms of the disease, they used antibiotics. Less than 20% of respondents let at least 2 days elapse from the onset of the symptom and then used antibiotics, an inappropriate practice as in the case of those who immediately used antibiotics. In terms of demographics, fewer significant associations were observed for practices regarding antibiotic consumption compared to those for knowledge and those for attitudes (Table 3). The LCA model describing antibiotic use practices consisted of two classes: class 1 representing the people with proper practices (80.1%), and class 2 encompassing the people with improper practices or uncertainty (19.9%), for which the data are represented graphically in Figure 3.

The multiple possible outcomes for the three questions in this scenario stem from the fact that people were not restricted to choosing a single answer from the possible choices listed in Table 5. People from the age group 35–44 years were half as likely (OR = 0.503; CI = 0.258–0.916; *p*-value = 0.032) to perform improper practices rather than proper practices, compared to people from the lowest age group (18–24 years). In terms of civil status, people in a relationship and single people were around twice as likely to perform improper practices, as opposed to proper practices, compared to married people. Finally, people with a level of income of RON 1501–3000 were more likely to perform improper practices regarding antibiotic consumption compared to people with the lowest level of income (>RON 1500), with no significant associations found for people with other categories of income level.

## 4. Discussions

Compared to the latest studies conducted in Romania [46,47,48] regarding the KAPs of antibiotic use and antibiotic resistance, this study reveals important information related to this topic, relevant throughout the country for all age groups and levels of education. The level of knowledge of Romanian respondents regarding the effectiveness of antibiotics is inadequate. About 50% of respondents believe that antibiotics are effective for dental pain (47.14%), fever (47.27%) or inflammation (52.31%). The effectiveness of antibiotics for dental pain has been agreed upon by a larger number of people in Lebanon, 71.9% [58], while in Jordan [62] and Malaysia [44], reported trial values were less than 30%. The confusion about whether or not antibiotics treat the majority of diseases was apparent in this survey, 24.31% of respondents being unsure about this subject. Our findings reported the confusion about the effectiveness of antibiotics on viruses, 37.51% of respondents believing that antibiotics treat viral infections. Even so, this percentage is lower than that reported in other European countries or in the United States [45,63]. The same confusion has been reported in previous studies [38,40,64,65]. In the present study, only 22.23% of the respondents knew that antibiotics kill bacteria, which is very worrying compared to the populations of other states, e.g., Malaysia, where, in a recent study [66], 49.0% of respondents knew that antibiotics kill bacteria. The difference is even greater when compared to the results obtained in Poland [67], where 80% of respondents knew that antibiotics kill bacteria. The knowledge about the side effects of antibiotics is considered satisfactory (81.77% of respondents agreed with the statement ″Antibiotics have side effects″). As in a recent Norwegian study [68], knowledge was considered satisfactory if ≥80% of the participants answered correctly for each statement. The present findings reveal that the knowledge of the general population in Romania regarding antibiotic resistance is quite low. Less than half of the study participants are aware of mechanisms for increasing microbial resistance or WHO predictions related to this aspect. In addition, the LCA analyses on knowledge regarding antibiotics efficacy, mode of action and antibiotic resistance showed that more than half of the Romanian population (53.3%) demonstrated less accurate or lack of knowledge related to this topic. The very low level of knowledge about antibiotic efficacy and antibiotic use in about 50% of our study respondents suggests an increased risk of misuse of antibiotics, which may lead to an increase in antibiotic resistance. Significant differences were found between the level of monthly income and the level of knowledge related to efficacy, mode of action, use of antibiotics and AMR. The results obtained show that people with higher incomes do not have better knowledge on this subject. The number of studies demonstrating the opposite results, where there was an association between a higher monthly income and a better level of knowledge, is limited [67,69]. As in many other studies, this study shows that a high level of education is associated with a better level of knowledge [37,39,44,49,58,69]. In Romania, the fact that education influenced the amount of knowledge, but income did not, could be attributed to the fact that education is highly subsidized by the state, up to the highest levels of postgraduate studies, which means that people from all economic and social backgrounds can freely attend higher education. Our study also found that older people appear to have more accurate knowledge than younger people. The need for adequate information for the correct approach to antibiotics is evident for the 18–24-year-old age group. They proved to have less accurate knowledge and responded in the highest percentage (69.19%, *n* = 1332). Similarly, the youngest age group responded in a larger proportion compared to other recent studies on the same subject [70]. In terms of attitudes toward antibiotic consumption, this study reveals that 52.73% of respondents try to have an antibiotic in the house, which can be determined by the concern and insecurity related to the COVID-19 crisis. A large percentage of respondents (83.64%) had an appropriate attitude disagreeing with the statement: ″If a family member feels unwell, I usually give them an antibiotic”. Our results identify 30.13% of respondents with improper attitudes toward antibiotic use when the symptoms are cold or flu. The results are similar to those obtained in other countries [62,65,69]. The attitudes toward antibiotic consumption differed according to some of the demographic characteristics. Females were less likely to have improper attitudes compared to males. A similar but stronger trend was noticed for people with higher levels of education, who were less likely to have proper attitudes compared to those with the lowest level of education. This result is contrary to other studies, where the high level of education is correlated with proper attitudes [39,59,61]. People aged 25–34 years were half as likely to have improper attitudes about antibiotic consumption, compared to younger people. Our results on gender and age influence are in accordance with numerous other studies [34,49,64]. Surprisingly, our findings reveal that people with higher levels of income (as well as an undisclosed income) are more likely to have improper attitudes towards antibiotic consumption than people with the lowest income level. The findings of this study reveal that 78.86% (*n* = 1518) of respondents use antibiotics only when prescribed by the doctor. A previous study reported that self-medication is a common practice in Romania among students [71], but the general population demonstrated a proper practice regarding this aspect. The results obtained are quite similar to those reported by other European studies, where 95% of the population stated that they use antibiotics only on when prescribed by the doctor [72]. The correct use of antibiotics and the continuation of the entire antibiotic treatment, even if the symptoms of the disease have disappeared, is one of the most important ways to reduce AMR. Unfortunately, 29.19% of respondents said that, when symptoms disappeared and they felt good, they stopped the antibiotic treatment, regardless of its duration. Various other studies have shown that other populations are less disciplined when it comes to completing antibiotic treatment. For example, 42.1% of Hong Kong respondents [73], 55.1% of the respondents from Western Saudi Arabia [70] or 51.5% of the Lebanese population [58] stopped their course of antibiotics if their symptoms improved. The best example was reported by Sweden [64], where less than 5% of the population would discontinue the treatment. Our findings reveal that individuals from the 35–44-year-old age group were half as likely to have improper practices than proper practices, compared to younger people. Single people were around twice as likely to have improper practices compared to married people. Moreover, in the present study, 80.1% of respondents showed proper practices for the use of antibiotics. As in other studies, there is no correlation between practices and knowledge regarding antibiotic use and antibiotic resistance [74,75,76]; but, overall, respondents who had less accurate knowledge were more likely to have improper attitudes toward antibiotic use.

## 5. Limitations

This study had several limitations. The main limitation of the study was the Internet access required to complete the questionnaire. Study participants received the questionnaire link via the Internet, which could have led to an overestimation of positive outcomes. Another limitation was that some key factors were not analyzed as covariates associated with the KAPs of antibiotic use and antibiotic resistance (such as employment status or location—rural/urban). A higher number of positive responses could also be due to the number of women who participated in the study, which was twice as large as that of men. Women are much more concerned with health than men. It is possible that respondents may have reported socially acceptable attitudes. In addition, it is possible that respondents reported more on their knowledge regarding the use of antibiotics and not their real attitudes and practices regarding the use of antibiotics. Another limitation was that our results were based on reported practices rather than measured practices. Despite these limitations, the present study provides important information for evaluating the KAPs of antibiotic use and antibiotic resistance among the general population of Romania. A strong point of our study is the large number of participants (*n* = 2067) and the high response rate (77%), as well as the fact that the respondents were not selected from certain educational units or certain sectors (hospital, pharmacies, clinics, etc.) in which they could have interacted with antibiotics or could have come across theoretical knowledge about antibiotics.

## 6. Conclusions

Our results show that the knowledge of the general public in Romania regarding antibiotics efficacy, mode of action and antibiotic resistance is considerably limited. A significant percentage of the population is quite confused about the effectiveness of antibiotics on viruses or bacteria. There is also confusion when it comes to the diseases that antibiotics treat. A fairly high percentage of the population found antibiotics to be the solution for most diseases. Unfortunately, the general population in Romania often associates antibiotics with toothache, flu or cough. Our study found that older people appear to have more accurate knowledge than younger people, but more frequent improper practices. The present study did not show a direct association between monthly income and proper practices, nor between monthly income and knowledge related to efficacy, mode of action, use of antibiotics and AMR. Keeping in mind the limitations of this study, its key findings may help policy makers in Romania to plan actions and strategies in the fight against AMR and mitigate the effects of this phenomenon. The results obtained can lead to the improvement of the KAPs of antibiotic use and antibiotic resistance through specific actions targeted towards the groups of Romanian populations where confusion, ambiguity or misconceptions have been reported.

## Figures and Tables

**Figure 1 ijerph-19-07263-f001:**
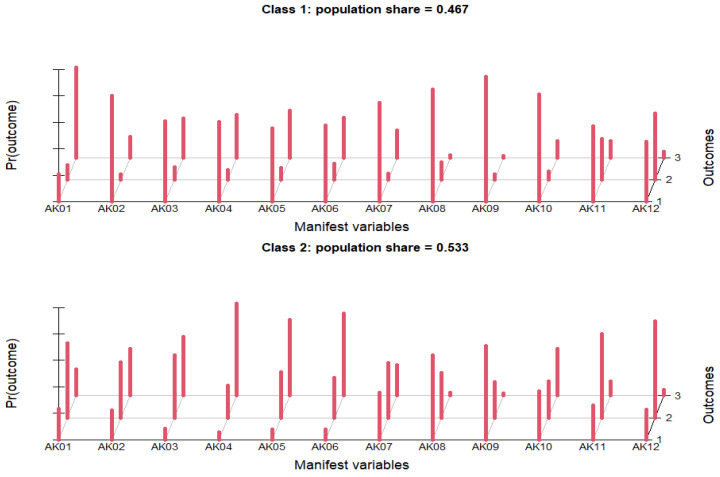
Knowledge regarding antibiotics efficacy, mode of action and antibiotic resistance.

**Figure 2 ijerph-19-07263-f002:**
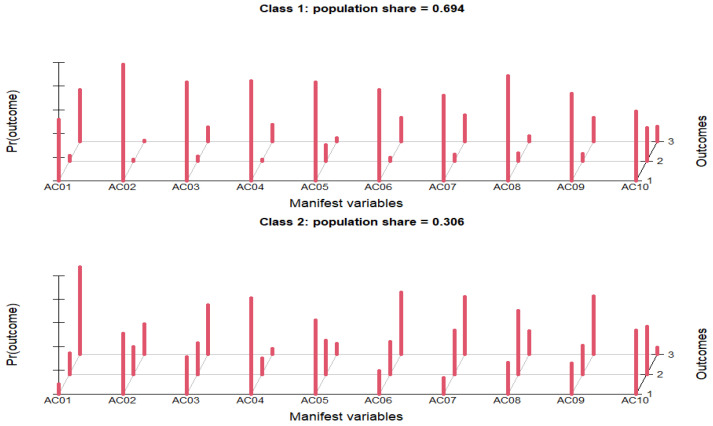
Attitudes towards antibiotic consumption.

**Figure 3 ijerph-19-07263-f003:**
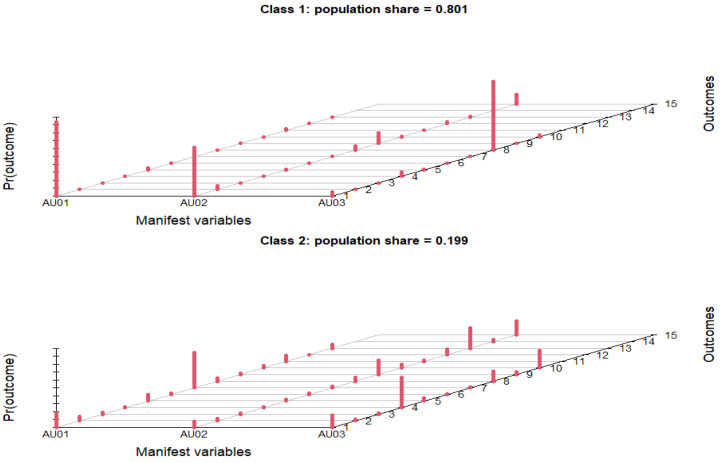
Respondents’ antibiotic use practices.

**Table 1 ijerph-19-07263-t001:** Demographic characteristics of the respondents.

Variables		N	%
Age (years)	18–24	1332	69.19
	25–34	301	15.64
	35–44	140	7.27
	45–54	124	6.44
	55–64	23	1.19
	≥60	5	0.26
Gender	Male	674	35.01
	Female	1251	64.99
Education level	Secondary	20	1.04
	High school	1296	67.32
	Graduate	454	23.58
	Postgraduate	155	8.05
Civil status	Married	309	16.05
	In a relationship	842	43.74
	Single	774	40.21
Monthly income	<RON 1500 (EUR 300) *	446	23.17
	RON 1501–3000 (EUR 301–600)	408	21.19
	RON 3001–4500 (EUR 601–900)	193	10.03
	RON 4500 (EUR 900)	164	8.52
	Undisclosed	714	37.09

* Exchange rate: 1 EURO = 5 RON.

**Table 2 ijerph-19-07263-t002:** Respondents’ knowledge related to antibiotics efficacy, mode of action and antibiotic resistance.

Statement		Agree	Unsure	Disagree
Antibiotics kill bacteria	AK01	22.23%	35.38%	42.39%
Antibiotics treat the majority of diseases	AK02	26.49%	24.31%	49.19%
Antibiotics treat viral infections	AK03	37.51%	29.61%	32.88%
Antibiotics reduce pain and inflammation	AK04	52.31%	16.52%	31.17%
Antibiotics significantly reduce fever	AK05	47.27%	22.55%	30.18%
Antibiotics help a lot with dental pain	AK06	47.12%	21.66%	31.22%
Excessive antibiotic use leads to AMR	AK07	53.87%	24.31%	21.82%
Antibiotics can cause allergies	AK08	73.77%	24.21%	2.03%
Antibiotics have side effects	AK09	81.77%	16.52%	1.71%
Antibiotic treatment may be discontinued when symptoms have disappeared	AK10	24.99%	17.66%	57.35%
Animal products can contain antibiotic residues, which may increase AMR when entering the human body	AK11	40.57%	47.95%	11.48%
The latest WHO prognoses estimate that, by 2050, antibiotic-resistant microorganisms will kill more people than cancer	AK12	33.3%	62.18%	4.52%

**Table 3 ijerph-19-07263-t003:** Multinomial logistic regression analysis of factors influencing KAPs of antibiotics use and antibiotics resistance among respondents.

Predictor Level		Knowledge (AK)			Attitudes (AC)			Practices (AU)		
		OR (Class 2/1)	CI	*p*	OR (Class 2/1)	CI	*p*	OR (Class 2/1)	CI	*p*
Age	18–24	1			1			1		
	25–34	0.736	0.545–0.993	0.045	0.489	0.344–0.689	<0.001	0.827	0.563–1.197	0.322
	35–44	0.436	0.283–0.668	<0.001	0.256	0.137–0.451	<0.001	0.503	0.258–0.916	0.032
	45–54	0.433	0.261–0.711	0.001	0.227	0.101–0.466	<0.001	0.898	0.459–1.694	0.746
	55–64	0.244	0.068–0.689	0.014	0.818	0.251–2.272	0.715	1.423	0.437–3.963	0.523
	>65	0.499	0.064–3.086	0.453	0.463	0.023–3.407	0.503	0	NA	0.972
Sex	Male	1			1			1		
	Female	0.57	0.465–0.699	<0.001	0.449	0.361–0.558	<0.001	0.919	0.717–1.182	0.508
Education	Secondary	1			1			1		
	High school	0.207	0.047–0.639	0.014	0.378	0.14–1.008	0.05	1.827	0.515–11.617	0.424
	Graduate	0.161	0.036–0.504	0.005	0.353	0.129–0.96	0.04	2.19	0.606–14.049	0.303
	Postgraduate	0.092	0.02–0.305	<0.001	0.34	0.112–1.018	0.053	2.031	0.515–13.594	0.373
Civil Status	Married	1			1			1		
	In a relationship	1.155	0.818–1.63	0.411	1.584	1.006–2.539	0.051	1.815	1.128–2.977	0.016
	Single	1.116	0.785–1.583	0.541	2.155	1.37–3.451	0.001	2.076	1.287–3.41	0.003
Income	<RON 1500	1			1			1		
	RON 1501–3000	1.356	1.02–1.804	0.036	1.6	1.162–2.204	0.004	1.52	1.07–2.163	0.02
	RON 3001–4500	1.279	0.875–1.875	0.205	1.813	1.172–2.79	0.007	1.17	0.709–1.899	0.53
	>RON 4500	1.205	0.786–1.85	0.394	1.756	1.064–2.875	0.026	1.528	0.897–2.566	0.113
	Undisclosed	1.137	0.889–1.454	0.307	1.65	1.256–2.175	<0.001	1.103	0.805–1.52	0.546

ORs: odds ratios; CIs: 95% confidence intervals; *p*: *p*-value.

**Table 4 ijerph-19-07263-t004:** Respondents’ attitude towards antibiotics consumption.

Statement		Agree	Unsure	Disagree
I always try to have an antibiotic in the house	AC01	52.73%	8.62%	38.65%
If a family member feels unwell, I usually give them an antibiotic	AC02	8.42%	7.95%	83.64%
If I have a few antibiotic pills left, I will use them with confidence for the following similar symptoms	AC03	21.19%	11.22%	67.58%
I use antibiotics according to the instructions for use	AC04	83.32%	5.14%	11.53%
Even if the shelf life has expired, antibiotics can still be used for up to one year	AC05	5.04%	18.18%	76.78%
I most often use antibiotics when the symptoms are cold/flu	AC06	30.13%	10.44%	59.43%
I most often use antibiotics when the symptoms are related to toothache	AC07	30.29%	15.48%	54.23%
I most often use antibiotics when the symptoms are related to gastrointestinal disorders	AC08	9.19%	21.19%	69.61%
When I feel better, I stop using antibiotics	AC09	29.19%	11.84%	58.96%
I need to pay close attention to the kind of food I eat when using antibiotics. There are many interactions between antibiotics and food consumed	AC10	57.3%	32%	10.7%

**Table 5 ijerph-19-07263-t005:** Respondents’ antibiotic use practices.

Statement		Frequency	Percentage
1. How do you use antibiotics?	AU01		
Only on the doctor’s prescription		1518	78.86
When I feel sick, with or without a doctor’s prescription		198	10.29
On the advice of the pharmacist		74	3.84
From the experience of acquaintances/friends		135	7.01
2. When you feel bad, the first impulse is:	AU02		
To go to a medical consultation		1036	53.82
To inform someone about your symptoms from the information on the Internet		576	29.92
To use antibiotics based on personal experience or knowledge		164	8.52
To ask the pharmacist for advice		149	7.74
3. How long after the onset of a disease symptom do you use antibiotics?	AU03		
Immediately, the passage of time does not help at all		200	10.39
1–2 days after the onset of symptoms		230	11.95
Only after I went to the doctor and received the prescription		1382	71.79
More than 4 days after the onset of symptoms		113	5.87

## Data Availability

Not applicable.

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
