# Peer review of "Knowledge, Attitudes and Practices Regarding Antibiotic Use and Antibiotic Resistance: A Latent Class Analysis of a Romanian Population"

_ijerph, 2022, doi:10.3390/ijerph19127263_

Round 1
Reviewer 1 Report
The manuscript is very well written by the author. The study conducted by the author will play a critical role in spreading awareness of antibiotic use and improving the quality of life of Romanian people.
Some of the mistakes I found the manuscript is as follows:
1. Correct the sentence on line 73-74.
2. Correct the spacing on line 80-81.
3. Correct reference no. 6.
Author Response
Dear Reviewer,
Thank you very much for your efforts during the manuscript revision, and for the valuable suggestions and comments.
Please find in the attached file our responses.
Have a great weekend.

Reviewer 2 Report
The article sheds light on an important aspect of the rise of antimicrobial resistance - knowledge, attitude and practices of the general public regarding antibiotic use. The strength of this study lies in the distribution of the population across different age groups, genders, education level, and income level. The questions categorised as knowledge and practices help survey the sample population well. The conclusions match well with the data the authors collected and may help enact policies to improve the knowledge and pactices regarding antibiotic use among the people of Romania.
However, the following points need to be corrected in order to make the article ready for publication:
-
Line 63-77 need extensive literature citations to support the claims in this section
-
Line 99: How is the questionnaire in this article different from the others published
-
Question AK02 “Antibiotics treat the majority of diseases” is ambiguous. As a result it is difficult to infer the answer. Line 186 suggests the correct answer to this question is correct, which is wrong unless it's clearly defined what “majority of diseases” includes.
Additionally, the writing can be improved to make the text more succinct.
Author Response
Dear Reviewer,
Thank you very much for your efforts during the manuscript revision, and for the valuable suggestions and comments.
Please, find in the attached file our responses.
Have a great weekend

Reviewer 3 Report
This study conducted a survey on antibiotic use and awareness of Romanian people, and it can be used as a reference data to establish the correct antibiotic use education and policies for Luminians.
However, the content of this paper is only an analysis of the results of the survey as a whole. Therefore, it feels more like a survey result report than a research paper.
In particular, in the discussion part, even though it was written in comparison with the cases of some other countries, the contents of the discussion and the result part were written very similarly, so it seems to have been written over and over again.
As a result of the survey, it seems meaningless to conclude that Romanians have a low knowledge, attitude and practice of antibiotic use. Therefore, in discussion, it is necessary to introduce various cases such as solutions to raise public awareness and policies or education of other success stories.
Therefore, in discussion part, additional preparation of good examples and various measures is required for each of at least three items, such as knowledge, attitude, and practice (KAP).
These suggestions will be a good reference for antibiotic resistance managers and the public to use antibiotics correctly and to reduce antibiotic resistance in the future.
Author Response
Dear Reviewer
Thank you very much for your efforts during the manuscript revision, and for the valuable suggestions and comments.
Please find in the attached file our response.
Have a great weekend.
All the best,
Ghimpeteanu et al.
